Novel immune cell infiltration-related biomarkers in atherosclerosis diagnosis

Dong Ruoyu 1
Li Jikuan 1
Jiang Guangwei 1
Han Ning 2
Zhang Yaochao 3
Shi Xiaoming 1 shixm2021@outlook.com
1 Department of Vascular Surgery, Hebei General Hospital , Shijiazhuang, Hebei , China
2 Department of Neurointervention, Hebei General Hospital , Shijiazhuang, Hebei , China
3 Department of Cardiothoracic Surgery, Cangzhou Central Hospital , Cangzhou, Hebei , China
Young Howard
Electronic publication date: 2023 May 1
Publication date: 2023
Volume: 11
Electronic Location ID: e15341
Received 2023 Feb 9; Accepted 2023 Apr 12
Copyright: © 2023 Dong et al.
Copyright year: 2023
Copyright holder: Dong et al.
License: This is an open access article distributed under the terms of the Creative Commons Attribution License, which permits unrestricted use, distribution, reproduction and adaptation in any medium and for any purpose provided that it is properly attributed. For attribution, the original author(s), title, publication source (PeerJ) and either DOI or URL of the article must be cited.
License URL: https://creativecommons.org/licenses/by/4.0/

Keywords: Immune cell infiltration, Atherosclerosis, WGCNA, Diagnosis

Funding: The authors received no funding for this work.

==============================
Background

Immune cell infiltration (ICI) has a close relationship with the progression of atherosclerosis (AS). Therefore, the current study was aimed to explore the role of genes related to ICI and to investigate potential mechanisms in AS.

Methods

Single-sample gene set enrichment analysis (ssGSEA) was applied to explore immune infiltration in AS and controls. Genes related to immune infitration were mined by weighted gene co-expression network analysis (WGCNA). The function of those genes were analyzed by enrichment analyses of the Kyoto Encyclopedia of Genes and Genomes (KEGG) and Gene Ontology (GO). The interactions among those genes were visualized in the protein-protein interaction (PPI) network, followed by identification of hub genes through Cytoscape software. A receiver operating characteristic (ROC) plot was generated to assess the performance of hub genes in AS diagnosis. The expressions of hub genes were measured by reverse transcription quantitative real-time PCR (RT-qPCR) in human leukemia monocyticcell line (THP-1) derived foam cells and macrophages, which mimic AS and control, respectively.

Results

We observed that the proportions of 27 immune cells were significantly elevated in AS. Subsequent integrative analyses of differential expression and WGCNA identified 99 immune cell-related differentially expressed genes (DEGs) between AS and control. Those DEGs were associated with tryptophan metabolism and extracellular matrix (ECM)-related functions. Moreover, by constructing the PPI network, we found 11 hub immune cell-related genes in AS. The expression pattern and receiver ROC analyses in two independent datasets showed that calsequestrin 2 (CASQ2), nexilin F-Actin binding protein (NEXN), matrix metallopeptidase 12 (MMP12), C-X-C motif chemokine ligand 10 (CXCL10), phospholamban (PLN), heme oxygenase 1 (HMOX1), ryanodine receptor 2 (RYR2), chitinase 3 like 1 (CHI3L1), matrix metallopeptidase 9 (MMP9), actin alpha cardiac muscle 1 (ACTC1) had good performance in distinguishing AS from control samples. Furthermore, those biomarkers were shown to be correlated with angiogenesis and immune checkpoints. In addition, we found 239 miRNAs and 47 transcription factor s (TFs), which may target those biomarkers and regulate their expressions. Finally, we found that RT-qPCR results were consistent with sequencing results.

Introduction

Atherosclerosis (AS) is the underlying cause of major adverse cardio- and cerebro-vascular events, such as stroke, peripheral artery disease and coronary artery disease, contributing to disability statistics and global death (Nong et al., 2022; Libby, 2021). The main pathogenic causes of AS include low-density lipoprotein (LDL) particle deposition in large- and medium-sized arteries, emigration of immune cells through damaged endothelial cells and the development of lipid plaques (Malekmohammad, Bezsonov & Rafieian-Kopaei, 2021). In addition, the interaction between lipid metabolism and immune response is also responsible for AS progression (Schaftenaar et al., 2016). Moreover, recent works have shown that anti-inflammatory interventions may be promising in the treatment of AS (Libby, 2021). Thus, identification of biomarkers related to immune and inflammation may provide theoretical and clinical guidance in AS prevention and treatment.

Immune cells are major players in immune system to mediate inflammation, and immune cell infiltration within vessel walls has close relationship with AS progression. By integrative analyses of CyTOF, CITE-seq and scRNA-seq, Fernandez et al. (2019) found multiple immune cell subpopulations, such as macrophages, monocytes and NK cells, in plaque and blood samples from AS patients. Furthermore, they found distinct features of T cells and macrophages in plaque samples with clinically symptomatic disease compared to asymptomatic disease (Fernandez et al., 2019). In mice, dendritic cells regulate T cell activation and adaptive immune responses to modulate atherogenesis (Subramanian et al., 2013; Daissormont et al., 2011). Using bioinformatics, Wang et al. (2022), Xia et al. (2021) and Xu, Chen & Yang (2022) found the proportions of immune cells were remarkably different between AS and control samples. However, so far to our knowledge, the role of genes related to immune infiltration in AS remains poorly understood.

Therefore, the current study is designed to give a more comprehensive mining of genes related to immune infiltration and evaluate their diagnostic potential in AS by bioinformatic strategies and in vitro validation. We hope our findings could facilitate the diagnosis and treatment for AS patients from immunological perspective.

Materials and Methods

Data source

In the current study, gene expression data from 32 AS plaque samples at stage IV and/or V lesions including core and shoulders of the plaque and 32 distant macroscopically intact control samples in GSE43292 (Ayari & Bricca, 2013) were used as the testing set to find diagnostic AS biomarkers. The demographic data of GSE43292 cohort have been reported by Ayari & Bricca (2013) in a previous literature. In addition, 29 atherosclerotic carotid artery samples and 12 healthy artery samples in GSE100927 (Steenman et al., 2018) were used as an external set to validate the expression patterns and diagnostic value of biomarkers identified in GSE43292. GSE43292 and GSE100927 were sourced from GEO database (http://www.ncbi.nlm.nih.gov/geo). The workflow of the current study was presented in Fig. 1.

Figure 1 The flow chart of the research.

DEG, differentially expressed genes; DIIC, differentially infiltrated immune cell.

Exploration of differentially expressed genes (DEGs) in GSE43292

Limma program in R was applied to mine DEGs using |logFC (fold change) | ≥ 1 and adjusted p-value < 0.05. The “ggplot2” package of R was used to create volcano plot. The heatmap was produced using “pheatmap” in R.

Identification of differentially infiltrated immune cells (DIICs) between AS and control

The proportions of IICs in AS and control samples was evaluated using the ssGSEA algorithm. This process was completed by using “GSVA” R package. The immune cells exhibiting significant differences between AS and control were examined using the Wilcoxon method with Benjamin & Hochberg adjusted p-value < 0.05.

Weighted co-expression network analysis

WGCNA was performed (Horvath & Dong, 2008) on all samples in the training set in order to screen the gene modules most associated with DIICs. To remove outliers, the hierarchical clustering trees for all samples was constructed, followed by the selection of optimal β value to build the scale-free network. Next, Pearson’s correlations between DIICs and gene modules were determined and presented in the heatmap. Finally, the most positive module and the most negative module correlated with DIICs were selected as key modules, and key moduler genes were used for the following analysis.

Mining and analysis of DEGs related to DIICs

To get DEGs related to DIICs, we overlapped DEGs with key modular genes identified in WGCNA and presented by a Venn diagram. Furthermore, KEGG pathway and GO analysis, comprising molecular function (MF), cellular composition (CC) and biological process (BP) were performed by “clusterProfiler” in R (Yu et al., 2012).

Identification of hub DIIC-related DEGs in AS

To identify hub DIIC-related DEGs in AS, a protein-protein interaction (PPI) network was first developed by uploading DIIC-related DEGs into STRING database (http://string-db.org) (Szklarczyk et al., 2015). Then Cytoscape software was applied for identifying core network by MCODE plug-in (Shannon et al., 2003). Genes in the core network were defined as hub DIIC-related DEGs in AS. Next, receiver operating curves (ROC) were plotted to assess the role of hub genes in AS diagnosis. If a hub gene has an area under ROC (AUC) > 0.7, then it was considered as a potential marker in AS diagnosis. In addition, their expressions and diagnostic potential of hub genes were validated in GSE100927. Then, the hub genes with consistent expression patterns and diagnostic potential were taken into the next analyses.

Characteristics and regulatory network of hub genes

To investigate the characteristics of hub genes and their relationships, we (1) analyzed their functional similarity by R package “GOSemsim”, (2) performed Spearman correlation analysis to determine whether their expressions were correlated, and (3) calculated angiogenesis and immune checkpoint scores using ssGSEA algorithm, followed by the calculation of correlations between hub gene expressions and angiogenesis/immune checkpoint scores. Furthermore, hub genes were imported into the miRNet database (https://www.mirnet.ca/) to search for miRNAs and transcription factors (TFs) that target and regulate the expressions of hub genes. Finally, the regulatory networks of miRNAs-hub genes and TFs-hub genes were developed using Cytoscape software.

Cell culture and RT-qPCR

To validate hub genes’ expressions, we purchased THP-1 cell line from the Cell Bank of the Chinese Academy of Sciences (Shanghai, China, CAS No: KG224). Cell culture were performed in RPMI-1640 medium (Sigma-Aldrich, St. Louis, MO, USA) containing 10% fetal bovine serum (FBS) in T25 flasks. At 37 °C and 5% CO2 THP-1 monocytes were induced to differentiate into macrophage by adding PMA at 100 ng/ml to the media for 48 h. Subsequently, the macrophages were incubated with 80 ug/ml ox-LDL for 24 h to transform into foam cells. Total RNA were extracted from foam cells and macrophages, respectively, using TRIzol Reagent (Thermal Fisher Scientific, Waltham, MA, USA). The purity and concentration of extracted RNA were detected for the following cDNA synthesis by SureScript-First-strand-cDNA-synthesis-kit (Servicebio, Guangzhou, China). Next, cDNA was applied for qPCR under 40 cycles at 95 °C for 60 s, 95 °C for 20 s, 55 °C for 20 s and 72 °C for 30 s using qPCR reagent from Servicebio (Wuhan, China). Gene expressions were analyzed by 2−ΔΔCt method. Primers of HMOX1, CHI3L1 and MMP9 were listed in Table 1.

Table 1 Primers used in the current study.

Genes	Forward	Reverse	
HMOX1	GAGACGGCTTCAAGCTGGTGA	CATGGCTGGTGTGTAGGGGAT	
CHI3L1	CCCTTGACCGCTTCCTCTG	CCTGGCTGGGCTTCCTTTAT	
MMP9	CAGAGATGCGTGGAGAGTCGA	AGGTGATGTTGTGGTGGTGCC	
GAPDH	CCCATCACCATCTTCCAGG	CATCACGCCACAGTTTCCC	

Results

102 DEGs and 27 DIICs were identified between AS and control

We discovered that 49 and 53 genes were significantly elevated and reduced, respectively, in AS than in controls in GSE43292 (Fig. 2A). The expressions of top 40 DEGs were presented in the heatmap (Fig. 2B). Meanwhile, ssGSEA algorithm was applied to estimate the infiltrations of 28 ICs in controls and AS as shown in Fig. 2C. After the Wilcoxon test, we observed that the average proportions of 28 ICs were much higher in AS group, and there were significant differences of ICs between AS and control except for type 2 T helper cells (Fig. 2D).

Figure 2 Identification of DEGs and DIICs between AS and control.

(A) The volcano map of the expressions of 102 DEGs discovered in GSE43292, with 49 up-regulated genes and 53 down-regulated genes. (B) The heatmap showing the expressions of top 20 up-regulated and top 20 down-regulated genes in GSE43292. (C) The heatmap of the infiltration levels of 28 immune cells in AS and control samples from GSE43292. (D) Wilcoxon test determines the differences of 28 immune cell infiltration levels between AS and control in GSE43292.

99 DIIC-related DEGs were screened by WGCNA

Next, we performed WGCNA to obtain DIIC-correlated modules. The sample clustering tree identified no outlier in the training set (Fig. 3A), followed by the generation of the sample dendrogram and trait heatmap (Fig. 3B). Afterwards, 10 is selected as the best soft threshold to develop the scale-free network (Fig. 3C). Finally, eight gene modules, including yellow, blue, gold, black, pink, blueviolet, turquoise and brown, were obtained (Fig. 3D). The correlations between DIICs and modules were calculated and presented in the heatmap (Fig. 3E). Accordingly, we selected most negatively (brown) and positively (pink) correlated modules with DIICs. Finally, a total of 4,568 genes in the pink and brown modules were screened as DIIC-related genes in AS. Thereafter, by overlapping 102 DEGs with 4,568 DIIC-related genes, we identified 99 DIIC-related DEGs (Fig. 3F).

Figure 3 Identification of DIIC-related DEGs.

(A) Sample clustering to detect outliers in GSE43292. (B) Sample dendrogram and trait map, in wich orange represents AS samples, and green represents control samples. (C) Selection of the best soft threshold to construct the scale-free network by WGCNA R package. (D) Cluster dendrogram, in which eight gene modules were obtained. (E) Heatmap showing the correlations between gene modules and DIICs. (F) Venn diagram showing 99 overlapped genes (DIIC-related DEGs).

By GO and KEGG pathway analyses, we identified 24 significantly enriched KEGG pathways, the top 10 of which were tryptophan metabolism, cAMP signaling pathway, long-term potentiation, PPAR signaling pathway, diabetic cardiomyopathy, complement and coagulation cascades, circadian entrainment, ECM-receptor interaction, Malaria and long-term depression (Figs. 4A and 4B). As for GO analysis, 52 BP, 11 MF and 21 CC were significantly enriched. As shown in Figs. 4C and 4D, the top 10 GO terms, such as tryptophan metabolic process, extracellular matrix disassembly, aromatic amino acid family metabolic process, and the corresponding DIIC-related DEGs involved in those GO terms, such as FABP4, MMP9, CXCL10 were presented.

Figure 4 Functional analysis of DIIC-related DEGs.

(A) Bubble chart and (B) Chord diagram of GO enrichment results. (C) Bubble chart and (D) Chord diagram of KEGG enrichment results.

10 hub genes could distinguish AS from control samples

The interactions among DIIC-related DEGs were explored in the STRING database (Fig. 5A) and were visualized by Cytoscape software (Fig. 5B). Then we screened core network from PPI network by MCODE plug-in (Fig. 5C). A total of 11 DIIC-related DEGs, including CASQ2, NEXN, MMP12, CXCL10, PLN, HMOX1, SELE, RYR2, CHI3L1, MMP9, ACTC1 were extracted from the core network (Fig. 5D). Among them, the expressions of MMP9, MMP12, HMOX1, CHI3L1, SELE and CXCL10 were upregulated, while the expressions of CASQ2, NEXN, RYR2, ACTC1 and PLN were down-regulated in AS samples compared to controls from training set (Fig. 6A). The ROC curves in the training set showed that those 11 genes had the potential to classify AS from controls (AUC > 0.7, Fig. 6B). Moreover, in the external dataset, the expression patterns and diagnostic value of CASQ2, NEXN, MMP12, CXCL10, PLN, HMOX1, RYR2, CHI3L1, MMP9, ACTC1 were validated (Figs. 6C and 6D). Thus, CASQ2, NEXN, MMP12, CXCL10, PLN, HMOX1, RYR2, CHI3L1, MMP9, ACTC1 were identified as hub DIIC-related DEGs and may act as AS diagnostic biomarkers.

Figure 5 Identification of hub DIIC-related genes in AS.

(A) The PPI network generated from SRING database. (B) After importing STRING network to Cytoscape software, the PPI network was regenerated by default parameters, in which red represents up-regulated genes, and blue represents down-regulated genes. (C) Hub module in PPI network is identified by MCODE plug-in. (D) MCODE score of hub genes.

Figure 6 Estimation of the diagnostic value of hub genes.

(A) Expression patterns of hub genes in the training set (GSE43292). (B) ROC analysis of hub genes in the training set (GSE43292). (C) Expression patterns of hub genes in the validation set (GSE100927). (D) ROC analysis of hub genes in the validation set (GSE100927).

Hub genes had close relationship with angiogenesis and immune checkpoints

Next, we explored the relationship among the hub genes. Correlation analysis showed that the expressions of them were highly correlated (Fig. 7A). Among them, MMP9 and CHI3L1 were most positively correlated (Fig. 7B), and NEXN and HMOX1 were most negatively correlated (Fig. 7C). Interestingly, we also found that there was a medium to high degree of similarity in their function (Fig. 7D). Moreover, in consideration the role of angiogenesis (Moulton, 2006; Perrotta et al., 2019) and immune checkpoints (Vuong et al., 2022; Alyagoob, Lahmann & Joner, 2020) in AS development, progression and treatment, we calculated the angiogenesis and immune checkpoints scores based on corresponding gene sets by ssGSEA algorithm. Further analyses showed that 10 hub genes were significantly correlated with angiogenesis score (Fig. 8A) and with immune checkpoints score (Fig. 8B).

Figure 7 The relationship among diagnostic biomarkers.

(A) Heatmap showing the correlations of expressions among biomarkers. (B) The most positively correlated biomarkers (MMP9 and CHI3L1). (C) The most negatively correlated biomarkers (NEXN and HMOX1). (D) Functional simalarity among biomarkers.

Figure 8 Potential molecular mechanisms and regulatory networks of biomarkers.

(A) Correlations between biomarkers and angiogenesis score. (B) Correlations between hub genes and immune checkpoint score. (C) MiRNA-biomarker regulatory network. (D) TF-biomarker regulatory network.

Hub genes were regulated by multiple miRNAs and TFs

Next, we investigated the miRNAs and TFs targeting and regulating the expressions of 10 hub genes. A total of 313 miRNA-hub gene pairs were searched to establish the miRNA-hub gene network, which includes 239 miRNAs and 10 hub genes (Fig. 8C). In the network, we found some hub genes were regulated by common miRNAs, for example that MMP9 and RYR2 were both regulated by mir-9-3p and mir-29b-3p. Meanwhile, 53 TF-hub gene pairs were obtained in miRNet database and a TF-hub gene network composed of six hub genes and 47 TFs was constructed (Fig. 8D). Also, we observed that two hub genes were regulated by the same TF, such as CHI3L1-STAT3-HMOX1, CHI3L1-SP1-MMP9, and CXCL10-RELA-MMP12.

RT-qPCR validation

Foam cells derived from macrophages may indicate the initial stages of AS (Chistiakov, Bobryshev & Orekhov, 2016; Maguire, Pearce & Xiao, 2019). Thus, in the current study, the foam cell model established from THP-1 monocytes was used, which was widely used in studying AS and AS-related disease (Yin et al., 2019; Huwait, Al-Saedi & Mirza, 2022; Mehta & Dhawan, 2020). After treatment with PMA, THP-1 cells were differentiated into macrophages and were defined as control group. Then THP-1 derived macrophages were treated with ox-LDL to form foam cells, which mimic the early stages of AS. Then we examined the expressions of HMOX1, CHI3L1 and MMP9 by RT-qPCR. The results showed that the expressions of HMOX1, CHI3L1 and MMP9 were significantly up-regulated in foam cells, which were consistent with the sequencing results (Fig. 9).

Figure 9 qPCR validation.

The MMP9 (A), CHI3L1 (B) and HMOX1 (C) mRNA levels were determined by qPCR in THP-1 derived macrophages without (NC group) and with ox-LDL (NC + ox-LDL group). The results were presented as mean ± S.D. Asterisks (****) indicate p-value < 0.0001.

Discussion

Identifying novel biomarkers and potential mechanisms is critical for treatments of AS patients. In the current study, in consideration of the importance of immune cell infiltration in AS pathology, we combined ssGSEA, WGCNA and differential expression analyses to comprehensively get immune cell-related genes involved in AS, and identified 10 genes with diagnosis potential for AS.

Firstly, we found that the proportions of 27 ICs were significantly elevated in AS, indicating that those immune cell types may contribute to AS. The roles of different macrophage and T cell subsets in AS have been reviewed by Anton and Goran that they function at different AS stage and affect plaque stability (Gistera & Hansson, 2017). In addition, innate immune response by dendritic cells and neutrophils are critical in AS initiation (Alberts-Grill et al., 2013). Thus, it is great significance to mine immune cell-related genes and investigated their role in AS. By WGCNA and differential expression analyses, we identified 99 DIIC-related DEGs, which were found to be involved in tryptophan metabolism and ECM-related biological processes through GO and KEGG pathway enrichment analyses. It has been reported that alteration of tryptophan metabolism plays an important role in AS. Briefly, inflammation in AS is driven by multiple cytokines, such as IFN-γ, which can up-regulate the expression of IDO. Then, tryptophan acts as a substrate for IDO and is degraded into kynurenine, leading to the progression of AS (Sudar-Milovanovic et al., 2022; Nitz, Lacy & Atzler, 2019; Wang et al., 2015). During this process, multiple innate and adaptive immune cells are involved, such as macrophages and dendritic cells (Nitz, Lacy & Atzler, 2019). Also, tryptophan metabolism is critical for the proliferation and function of T cells (Fallarino et al., 2006; Mezrich et al., 2010; Munn et al., 2005). As for vascular ECM, it is mainly composed of elastin, microfibrils, collagens, proteoglycans and other glycoproteins (Ma et al., 2020). Degradation of elastin into elastokines promotes AS by regulating uptake of ox-LDL (Kawecki et al., 2019), remodulating function of macrophages (Hsu, Tintut & Demer, 2021) and promoting angiogenesis (Heinz, 2020). Collagens have been reported to have an essential role in determining the stability of AS plaque (Xu & Shi, 2014; Johnston, Gaul & Lally, 2021). Proteoglycans and their GAGs are regulators in lipid retention, activation of immune response and proliferation of smooth cells, which contribute to AS progression (Viola et al., 2016). Those reports and our findings suggest that the identified DIIC-related DEGs may regulate AS at least by tryptophan metabolism and ECM.

From those 99 DIIC-related DEGs, we selected 10 potential markers with good performance distinguishing AS from control by constructing PPI network and performing ROC analysis. CASQ is the most abundant Ca2+-binding protein in skeletal and cardiac muscle sarcoplasmic reticulum. The CASQ2 and CASQ1 genes have been found to be mutated in patients with catecholamine-induced polymorphic ventricular tachycardia (Lodola et al., 2016), which could lead to sudden death (Rajagopalan & Pollanen, 2016). The PLN plays an important role in regulating sarcoplasmic reticulum (SR) function as well as cardiac contractility. Perisic Matic et al. (2016) found that PLN expression was sharply decreased in smooth muscle cells treated with IFN-γ and ox-LDL, which mimic the environment of AS, and polymorphism in PLN was found to be associated with maximum common carotid artery thickness. Through mitochondrial function, calcium homeostasis and excitation-contraction coupling, RYR2 activity in myocytes is associated with electrical and contractile dysfunction in the arrhythmogenesis heart of aged human (Hamilton & Terentyev, 2019). The expression of CXCL10 was observed in different stages of AS lesion development (Mach et al., 1999), and knockout of ApoE and CXCL10 in mice lead to significantly smaller lesions, less CD4+ T cells and increased regulatory T cells compared to ApoE−/− mice (Heller et al., 2006). As for HMOX1, its expression was significantly elevated in high-fat diet ApoE−/− mice, and knock-down of HMOX1 in endothelial cells impaired overload of Fe2+, ROS and lipid peroxidation, which led to impaired ferroptosis and may attenuate diabetic AS development (Meng et al., 2021). Hu et al. (2019) found down-regulation of NEXN expression in AS plaques compared with that in healthy artery tissues. Further in vivo experiments showed that NEXN+/− ApoE−/− mice had more AS lesions and increased expressions of ICAM1, IL6, TNFα and MMP9 et al. compared to NEXN+/+ ApoE−/− mice (Hu et al., 2019), indicating a protective role of NEXN against AS. As for CHI3L1, it is a risk factor of AS, in which its expression is up-regulated and knockdown of CHI3L1 reduced lipids, macrophages and expressions of local proinflammatory mediators in plaques (Gong et al., 2014). Higher MMP9 and MMP12 expressions were observed in AS patients (Gong et al., 2014; Marcos-Jubilar et al., 2021), which is consistent with the public sequencing results. Nuciferine exerts its protective role against AS by Calm4/MMP12/AKT signaling to regulate the migration and proliferation of vascular smooth muscle cells. Although those genes have been reported in AS and AS-related diseases, the exact mechanisms of those genes in regulating AS remains poorly understood.

At last, RT-qPCR was applied to examine the expression patterns of MMP9, CHI3L1 and HMOX1 via THP-1 cellular model. We found that their expressions were consistent with what we observed in sequencing data. But it is needed to notice that the in vitro cellular model can’t fully represent what is going on in vivo in consideration of its simple cellular environment. It has been reported that cell-cell communications and cell-cell interactions such as B cell-T cell interaction (Ma et al., 2022), cross-talk among endothelial cells, immune cells and vascular smooth muscle cells (Sorokin et al., 2020) and even the structure of extracellular matrix (Halabi & Kozel, 2020), are important for the development of AS plaques. Thus, in vivo plaque and control samples are still needed to further verify their expressions.

There are some limitations in our study. First, the expressions of identified biomarkers need to be determined in AS and control tissues. Second, how they interact with immune cells and how they regulate AS should be elucidated in in vitro and in vivo experiments. Last but not the least, their potential to act as biomarkers in AS treatment and therapeutic targets in AS treatments needs to be verified in real clinic world. Thus, in future work, we plan to explore the underlying mechanisms of those genes in the regulation AS, including but not limited to: (1) examine the expression patterns of those genes in AS patients under different stages by western blotting or qPCR, (2) explore the immune cell infiltration in AS using mice AS model via and detect whether those biomarkers are colocalize with infiltrated immune cells by immunofluorescence, (3) select 2–3 most interested genes and construct crispr knock-out and/or overexpression via lentivirus infection to monitor its role in the formation of AS and inflammation, and (4) examine its effect on angiogenesis by tube formation assay in vitro.

Conclusions

For the first time, we identified 10 DIIC-related DEGs, including CASQ2, PLN, RYR, CHI3LI, NEXN, MMP9, MMP12, HMOX1, CXCL10, and ACTC1 in AS. These 10 genes were significantly correlated with angiogenesis and immune checkpoints. Those findings provide future directions in unveiling the molecular mechanisms of AS and also offer novel potential biomarkers and therapeutic targets for AS patients.

Supplemental Information

Supplemental Information 1 Raw data of RT-qPCR.

The raw CT values and processed result of RT-qPCR.

Click here for additional data file.

We are grateful for authors who share their sequencing data on the GEO database.

Additional Information and Declarations

Competing Interests

Author Contributions

Data Availability

The authors declare that they have no competing interests.

Ruoyu Dong conceived and designed the experiments, performed the experiments, analyzed the data, prepared figures and/or tables, authored or reviewed drafts of the article, and approved the final draft.

Jikuan Li conceived and designed the experiments, performed the experiments, prepared figures and/or tables, authored or reviewed drafts of the article, and approved the final draft.

Guangwei Jiang analyzed the data, prepared figures and/or tables, authored or reviewed drafts of the article, and approved the final draft.

Ning Han analyzed the data, prepared figures and/or tables, authored or reviewed drafts of the article, and approved the final draft.

Yaochao Zhang performed the experiments, authored or reviewed drafts of the article, and approved the final draft.

Xiaoming Shi conceived and designed the experiments, authored or reviewed drafts of the article, and approved the final draft.

The following information was supplied regarding data availability:

The raw data are available in the Supplemental File.

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
