# Peer review of "Novel immune cell infiltration-related biomarkers in atherosclerosis diagnosis"

_PeerJ, doi:10.7717/peerj.15341_

## Round 0.1 · original submission · Minor Revisions

I am pleased to inform you that the reviewers have recommended acceptance of your manuscript pending minor revisions. Please address the minor issues raised by reviewer #1 and resubmit your manuscript. In your response please be sure to explain how you have addressed the recommendations for revision. Thank you for choosing PeerJ for publication of your data and we look forward to receiving the revised manuscript.

Reviewer 1 ·

Basic reporting

Overall, the manuscript is clearly written and all methodologies are well explained and detailed.
The overarching goal of the investigation was to define markers of immune infiltration of atherosclerotic plaques as immune cells are appreciated to be key drivers of the progression of atherosclerosis. The authors provide ample detail on the approach to mine data already available to pinpoint protein-protein interactions that may be important, and ultimately identified immune cells and markers (immune cell-related genes) that correlated with atherosclerosis. Expression of specific genes were found to distinguish healthy from disease plaques and THP-1 cells, exposed to oxidized-LDL, recapitulated several of the findings.

Comment/Suggestion:
The first Figure includes a flow chart describing the analyses. One recommendation would be to provide more information and spell out acronyms in this flow chart. It would be more useful and a nice reference guide for readers to have a detailed and comprehensive workflow. In its current form, it requires flipping to the text to follow the acronyms.

Experimental design

This is a thorough investigation of publicly available transcriptional data supporting the role of immune cell infiltration in atheroprogression. Data sets analyzed included atherosclerotic patient samples and healthy controls. The approach taken to identify markers of immune infiltration used weighted gene co-expression network analysis, gene ontology and KEGG pathway enrichment analysis, followed by constructing a protein-protein interaction network. Cytoscape software was used to identify hub genes.
Finally, quantitative RT-PCR experiments were performed using THP-1 cells, exposed to oxidized lipids, in order to validate the findings.

Comment/Suggestion: While thorough and easy to follow, it would benefit the reader to provide details on the type of plaque analyzed and important demographic data about disease.

Validity of the findings

Key outcomes from the study were the identification of specific genes that are up or down regulated in atherosclerosis. Whether and how this information could be used diagnostically is not clear, however, the data are compelling in the sense that they define specific factors that are associated with disease progression and may help identify targets for mitigating disease. It would be interesting for the authors to discuss how this information could be used.

Additional comments

1. The authors find some differences with respect to their observations from human patient samples and their in vitro THP-1 studies. Comparing and contrasting the two systems (in vivo conditions where multiple cell types are present versus in vitro where just one cell type is present) may provide a discussion point about what cell-cell interactions are important for disease progression.

2. There was one point in the discussion that seemed somewhat paradoxical. Line 266-268-- impaired ferroptosis promoted atherosclerosis-- perhaps the authors can discuss this in more detail?

Reviewer 2 ·

Basic reporting

'no comment'

Experimental design

'no comment'

Validity of the findings

'no comment'

Additional comments

The paper by Ruoyu Dong et al. is worthy of acceptance.

·

Basic reporting

The article is written using clear, unambiguous,technically correct text. Relevant prior literature were appropriately referenced. the artical structure,figures, tables,raw data were appropriately handled.

Experimental design

no comment

Validity of the findings

the results are reliable, robust, statistically sound, and controlled. Conclusions are well stated.

Additional comments

no comment

---

## Round 0.2 · accepted · Accept

Thank you for addressing the concerns raised by the reviewer. I am pleased to inform you that your manuscript is now acceptable for publication. Thank you for choosing PeerJ.